# Clinical Characteristics and Disease Course of Fibrosing Interstitial Lung Disease Patients in a Real-World Setting

**DOI:** 10.3390/medicina59020281

**Published:** 2023-01-31

**Authors:** Maritta Kilpeläinen, Tuuli Hirvonen, Katariina Perkonoja, Samuli Hirsjärvi

**Affiliations:** 1Department of Pulmonary Diseases, Turku University Hospital and University of Turku, 20500 Turku, Finland; 2Auria Clinical Informatics, Turku University Hospital, Hospital District of Southwest Finland, 20521 Turku, Finland; 3Boehringer Ingelheim Finland Ky, 00180 Helsinki, Finland

**Keywords:** real-world data, interstitial lung disease, idiopathic pulmonary fibrosis, progressive pulmonary fibrosis, incidence, prevalence

## Abstract

*Background and Objectives*: This single-center retrospective study was conducted to describe clinical characteristics and the disease course of patients with interstitial lung diseases (ILD). *Materials and Methods*: The study included adult patients with fibrosing ILD (IPF, non-IPF fibrosing ILD (F-ILD), and non-IPF progressive pulmonary fibrosis (PPF)) treated between 2014 and 2017. Patients were followed annually from the first visit until the end of the study period in 2019. Data were collected from the Turku University Hospital data lake and analyzed using descriptive statistics. *Results*: 591 patients formed the patient cohort: 110 had IPF, 194 F-ILD, 142 PPF, and the remaining 145 patients were uncertain, F-ILD-U, whose disease progression nature could not be confirmed by FVC measurements. There were more males in each patient group and median age of the groups was similar, although there were younger patients in the PPF, F-ILD, and F-ILD-U groups. PPF patients had more UIP pattern than F-ILD patients. Exposure-related ILDs were clearly the most found ILD diagnoses for both PPF and F-ILD, followed by unclassifiable IIP. Baseline FVC % predicted reduction in every group was moderate. Half of the patients in each group had comorbidities, and the most common were cardiovascular diseases, diabetes, sleep apnea, and chronic lower respiratory diseases; F-ILD-U patients had malignant diseases as well. IPF patients had less medications than the other groups. Glucocorticoids were the most used medication in all patient groups. More PPF and F-ILD patients remained in the follow-up than IPF and F-ILD-U patients. Similarly, mortality of F-ILD-U was the highest, followed by IPF. Evolvement of lung function, oxygen use, and number of acute hospitalizations were similar for IPF and PPF patients whereas the corresponding results were always better for F-ILD patients. *Conclusions*: The disease course of IPF and PPF was similar, and PPF patient amount exceeded the amount of IPF patients.

## 1. Introduction

Interstitial lung diseases (ILDs) comprise a heterogeneous group of several rare pulmonary disorders characterized by diffuse interstitial (or parenchymal) inflammation, fibrosis, or a combination of both [1].

Among ILDs, there is a subgroup of patients that, regardless of the underlying condition, display a progressive pattern that is self-sustained and continuously worsens over time. Traditionally, these have be pooled under the umbrella term of “progressive fibrosing ILD” (PPF) [2,3]. Recently, the term “progressive pulmonary fibrosis” (PPF) was introduced to describe progressive ILDs other than IPF [4]. PPF is used in the current study for these patients. IPF, which is a progressive ILD, is one of the most common and most studied ILD [5]. Other types of ILD that are most likely to be progressive are idiopathic nonspecific interstitial pneumonia (iNSIP), unclassifiable idiopathic interstitial pneumonias (uIIP), interstitial pneumonia with autoimmune features, RA-ILD (rheumatoid arthritis), SSc-ILD (systemic sclerosis), HP (hypersensitivity pneumonitis), sarcoidosis, and ILDs related to other occupational exposures [6]. The natural history of PPF is equal to IPF, with a worsening of respiratory symptoms, lung function, quality of life, and functional status, as well as early mortality [2,3]. For example, patients included in the placebo group of the INBUILD PF-ILD trial had similar clinical behavior in terms of FVC decline and mortality compared to the placebo groups of the INPULSIS IPF trials [7].

The prognosis is variable, being worst in IPF (approximately 20–30% of all ILDs), with a median survival of only 2.5–3.5 years [2]. Both the presence (of fibrosis) and progression of fibrosis and lung function impairment over time (progressive disease) are predictors of worse clinical outcomes. Therefore, fibrosing ILDs other than IPF may express a similar poor outcome. Patients with IPF frequently experience diagnostic delays that can negatively affect their prognosis [8]. With the introduction of antifibrotics to treatment selection, this also applies to the other progressive ILDs. To receive an early treatment intervention, it is essential to identify these patients. Understanding of clinical characteristics together with disease course is an integral part of the treatment pathway.

Despite the poor outcome of these fibrosing lung diseases, availability of epidemiological data across different countries has been limited. Additionally, clinical course of disease of these patients has not been widely evaluated. The PERSEIDS study evaluated the prevalence and incidence of ILD in mid-sized European countries [9]. Finland (particularly the Hospital District of Southwest Finland (HDSF)) was one of the study sites. For Finland, the overall ILD prevalence (per 105 people) was 99–202 and incidence 26–54. IPF prevalence and incidence were 13–27 and 3–7, and PF-ILD prevalence and incidence were 16 and 4, respectively.

To build up on the epidemiological findings of the PERSEIDS study, the current study investigated characteristics of patients with different fibrosing ILDs, i.e., IPF, non-IPF fibrosing ILD (F-ILD), and non-IPF progressive pulmonary fibrosis (PPF) patients. Patients whose fibrosis progression status could not be confirmed (F-ILD-U, “uncertain”) were handled as a separate group. Course of disease in all those patient groups was followed as well.

## 2. Materials and Methods

### 2.1. Study Population and Data Collection

This single-center, retrospective, observational cohort study included all adult patients (≥18 years of age) with fibrosing ILD who received treatment at the HDSF between 1 January 2014 and 31 December 2017. IPF and non-IPF ILD-diagnosis were confirmed by study physicians (MK, TH) from the electronic medical records (EMRs). Patients were followed from the first visit in 2014–2017 and referred to as baseline until the end of the study period (31 December 2019) or death, whichever came first. Data were collected retrospectively from the Turku University Hospital data lake.

The data collected included age, gender, UIP (usual interstitial pneumonia) pattern, fibrosing ILD diagnosis, time since the fibrosing ILD diagnosis, smoking status, comorbidities (ICD-10), selected medication, body mass index (BMI), lung function (FVC (forced vital capacity), TLC (total lung capacity), DLCO (diffusing capacity for carbon monoxide)), 6 min walking test distance (6MWT) and lowest oxygen saturation (SpO2), number of acute hospitalizations, oxygen usage, change in fibrosis, and date of death. Detailed descriptions of ILD diagnoses, comorbidities, medications, and acute hospitalizations are presented in Appendix A.

### 2.2. Outcome Measures

A patient with non-IPF fibrosing ILD (F-ILD) was considered to have a progressive disease (PPF) if any of the following were true based on patient’s EMRs: (1) a relative decline of ≥10% in FVC % predicted within 2 years, or (2) a relative decline of between 5 and 10% in FVC % predicted within 2 years, and ≥1 ILD hospitalization (not including emergency room visits) within 2 years (after ILD diagnosis), or increasing extent of fibrosis on high-resolution computed tomography (HRCT) within 2 years, or start or increase in use of oxygen within 2 years, or death due to respiratory event within 2 years.

Normal FVC predicted was calculated by using the Global Lung Function 2012 equations [10]. The change in fibrosis was confirmed by study physicians from the HRCT statements. Both TLC and DLCO predicted normal were calculated using Global Lung Function Initiative reference values [11,12].

A patient with non-IPF fibrosing ILD was considered to have a UIP pattern if the HRCT results showed any of the following: (1) definite honeycomb lung destruction with basal and peripheral predominance in absence of atypical features (specifically: nodules and consolidation; ground glass opacity, if present, is less extensive than reticular opacity pattern); or (2) presence of reticular abnormality and traction bronchiectasis consistent with fibrosis with basal and peripheral predominance in absence of atypical features. The existence of a UIP pattern was confirmed by study physicians from the HRCT statements.

Smoking status was extracted from the EMRs by using an ULMFit NLP model trained for this specific purpose by Auria Biobank [13]. The percentage from theoretical distance of 6MWT and lowest SpO2 were extracted from the EMRs by using regular expressions as presented in Appendix A.

Time since ILD diagnosis was calculated retrospectively by searching the EMRs for the first ICD-10 diagnosis date corresponding to the diagnosis confirmed in the study and calculating the difference between this and the baseline date.

### 2.3. Statistical Analyses

For each patient included in the study, the first date of visit between 2014 and 2017 was defined as the baseline. The closest observation to that date within each follow-up year was selected as the follow-up point to depict annual monitoring. Patients who did not have enough FVC measurements to confirm the progression status of fibrosis were included as a separate group in the analyses (F-ILD-U).

Demographics, comorbidities, and medication were presented at baseline and other outcome measures were presented from the first available observation onwards. The continuous variables in the data were described using medians along with lower and upper quartiles (Q1, Q3) and/or range (min–max). Categorical variables were presented using observed frequencies and proportions. The findings were further illustrated using a cumulative incidence curve (CIC) and bar and line charts.

All available data were used in the analyses and missing values were described and used without imputations. Statistical analysis and tables and figures were produced with R version 3.6.3 (R Core Team, Vienna, Austria, 2018) using RStudio Server in Auria Clinical Informatics’ secure operating environment.

## 3. Results

### 3.1. Baseline Characteristics

A total of 591 patients formed the patient cohort (Figure 1). A total of 110 patients (19%) had IPF and 142 (24%) had PPF. A total of 194 patients (33%) had F-ILD, and the remaining 145 patients (25%) formed a group F-ILD-U (uncertain) whose disease progression nature could not be confirmed due to an insufficient amount of FVC measurements. Major diagnoses in the group of patients with no fibrosis (*n* = 481) were sarcoidosis (*n* = 255) and mixed connective tissue disease (*n* = 79). Most common diagnoses in the no ILD group (*n* = 66) were asbestos plaque disease (*n* = 21), asthma (*n* = 11) and sleep apnea (*n* = 10). Diagnosis and/or fibrosis could not be confirmed (*n* = 94) if the medical chart did not mention fibrosis and/or the diagnosis was not confirmed.

There were more males in each patient group, about 65–68% (Table 1). Median age of F-ILD patients was the youngest (70 vs. 73–75 years of the other groups). There were younger patients in PPF, F-ILD, and F-ILD-U groups (min. age 20–24 years) as compared to IPF (min. age 54 years). PPF patients had more UIP pattern than F-ILD patients (32.4 vs. 26.8.%). Median time from initial presentation, since the first ILD diagnosis, was 24.5 months for PPF patients and longer, 35.0 months, for F-ILD patients. IPF patients were diagnosed without delay (zero months). IPF patients were more often never-smokers (28.2%) than PPF (16.9%) and F-ILD (17.0%) patients. Exposure-related ILDs were clearly the most found ILD diagnoses in the cohort for both PPF and F-ILD, about 39–42% of the respective patients. The next PPF diagnoses were unclassifiable IIP, NSIP and SSc-ILD, and NSIP, unclassifiable IIP and RA-ILD in F-ILD.

Patients whose fibrosis progression status could not be confirmed (*n* = 145) are presented as a separate group (F-ILD-U). With confirmed fibrosis status, these patients would belong either to the PPF or F-ILD group. Compared to the two better-defined fibrosing groups (PPF and F-ILD), these patients were older (median age 75 vs. 73 and 70 years) and were mostly men, and there were more never-smokers (26.9%). They had even more UIP pattern than patients in the PPF group (35.2 vs. 32.4). Similar to IPF patients, there were no delay in the diagnosis (median zero months since the first ILD diagnosis). Unclassifiable IIP (33.8%) and exposure-related ILDs (31.0%) were the most common diagnoses, followed by NSIP.

Baseline lung function (FVC, DLCO, TLC) refers to the first observed value in the patient’s record during the study period. FVC was the most available lung function measurement in the data lake (least “missing” status). A total of 34.5% of the F-ILD-U patients did not have any FVC measurements; the rest of this patient group had one registered reading. Baseline FVC % predicted reduction in every group was moderate, ranging from 77% (IPF) to 80% (PPF). IPF patients had moderate DLCO % predicted reduction (55%) whereas the decrease in the other groups was mild (about 59–64%). TLC % predicted reduction in all groups was mild, and lowest in IPF (74% vs. about 79%). The lowest oxygen saturation measurement (about 84–88% in all groups) and 6MWT (about 67–82% from theoretical distance in all groups) were reported only to about 10–20% of patients regardless of the group, except for 45.5% of the IPF patients.

A little more than a half of the patients in each group had comorbidities as categorized “any comorbidity” (Table 2). The most common comorbidities of IPF patients were cardiovascular diseases followed by diabetes and sleep apnea. The most frequent comorbidities in PPF, F-ILD, and F-ILD-U were similar: chronic lower respiratory diseases, diabetes, and cardiovascular diseases. In addition, F-ILD-U patients had more malignant diseases than the other groups. A vast majority of patients in every group were not obese (BMI ≤ 30).

IPF patients had less medications (31.8%) than the other ILD patients distributed in the PPF (41.5%), F-ILD-U (39.3%), and F-ILD (37.6%) groups (Table 2). Glucocorticoids were the most used medication in all patient groups, followed by immunosuppressants (azathioprine/methotrexate).

### 3.2. Disease Course

The disease course of each patient was followed for a maximum of five years. The reported follow-up years were always in relation to the observed baseline situation of each patient. The proportions of patients still in follow-up at each year are presented in Figure 2. The proportions of PPF and F-ILD patients in the follow-up remained higher throughout the years than the proportion of IPF patients which decreased faster. Clearly, less patients in the F-ILD-U group were in the follow-up each year when compared to the other patients. Similarly, the mortality of the patients in the F-ILD-U groups was the highest with a median overall survival (mOS) of 29.8 months (Figure 3). The mOS in the IPF group was 47.4 months. The mOS times for the PPF and F-ILD groups were not reached during the five-year follow-up, yet the F-ILD group showed the best survival.

The possible evolvement of fibrosis status was evaluated from HRCT statements. A mention of increased fibrosis was found for 21.8, 17.8, 19, 16.3, and 6.5% of IPF patients in the follow-up years 1 to 5, respectively. A proportion of 7 and 7.7% of PPF patients had a mention of increased fibrosis at years 1 and 2, respectively. For the rest of the groups and years, less than 5% were estimated to have increased fibrosis. Decreased fibrosis was reported for 1–3 patients yearly until the fourth year in the F-ILD group. Apart from these, information about an increase or a decrease was missing in the HRCT statements (fibrosis status data are presented in Appendix A).

Not all the patients had baseline lung function measurements available. Therefore, the first observed lung function reading was determined as a baseline, followed by yearly follow-up measurements. However, e.g., 79.4% of the IPF patients had a baseline FVC measurement from the first visit during the observation time of this study. The number of patients yearly in each group are presented in Appendix A. Progression of lung function measurements are presented in Figure 4. FVC of IPF and PPF patients worsened until the follow-up year 3, from 78% to 66% and 72% to 70% predicted, respectively, but was higher for the patients still in follow-up at years 4 and 5. FVC of F-ILD was higher, about 80–83%, during the follow-up, with a peak 88% at the year 3. DLCO of IPF and PPF patients was similar and constant during the follow-up—about 50–57% predicted. Percentages of F-ILD patients were higher and increased from 63% to 70% predicted at the end for those still in follow-up. There was more variation in the F-ILD-U during the follow-up. TLC of IPF and PPF patients was also similar and constant—about 70–75% predicted. Values in the F-ILD and F-ILD-U groups were higher, about 79–85% predicted throughout the follow-up.

Oxygen use was substantial and similar in the IPF and PPF groups until the fourth follow-up year, in about 14–18% of the patients, with a drop in the use for the patients in the follow-up at the fifth year (Figure 5). A lower proportion, about 5–9% of the patients in the F-ILD group, received oxygen therapy. Oxygen use of the F-ILD-U group was about 12–14% in the first two years, and about 5–6% in the following two years. IPF patients had more acute hospitalizations than PPF patients at the first year (22.7% vs. 14.8% of the patients) but during the following four years, the proportion of patients with a hospitalization event in these groups was similar—about 11–20% yearly (Figure 5). F-ILD patients had less acute hospitalizations—about 3–11% yearly. Clearly, the greatest proportion of patients with acute hospitalization events was in the F-ILD-U group at the first follow-up year—38.4%. Thereafter, their proportion was lower and comparable to the other groups. The median amount of acute hospitalizations/patient yearly was one in every patient group and in each follow-up year.

Follow-up 6MWT and the lowest SpO2 were reported for few patients only: 18 IPF patients had median (Q1, Q3) 6MWT 83.0% (60.5, 93.8) (from theoretical distance) and the lowest SpO2 81.5% (79.0, 87.0) in the first year, whereas the results for eight patients in the second year were 76.0% (47.5, 99.0) and 82.5% (79.5, 87.3), and for three patients in the third year were 93.0% (75.0, 105) and 88.0% (87.0, 91.0), respectively. In the PPF group, eight patients were followed-up in the first year with a median 6MWTD 54.0% (50.5, 64.5) and the lowest SpO2 84.0% (74.0, 87.5), four patients in the second year had 53.0% (48.8, 61.3) and 86.0% (81.0, 89.5), respectively, and one patient in the third year. The median 6MWTD and lowest SpO2 for 5 F-ILD patients in the first year were 77.0% (57.0, 96.3) and 87.0% (84.0, 89.0), and in the second year 62.0% (50.0, 66.0) and 81.0% (77.0, 89.0), respectively. Measurements in the third year for three patients were 61.0% (56.0, 66.0) and 83.0% (78.5, 86.0), respectively. In the F-ILD-U group, two patients had measurements in the first year and one in the second year.

The median BMI of IPF and PPF patients decreased with a similar trend during the follow-up (Figure 6). The median BMI of the F-ILD group remained higher and somewhat constant than values of the two former groups. Instead, the BMI in the F-ILD-U group was clearly lower during the first two years but increased for the patients who were still in the follow-up from the third year onward. Patients were generally overweight but not obese.

## 4. Discussion

Baseline characteristics of the patients in this study were well in line with previous RWE findings.

Based on the estimated population of the study area (470,000 inhabitants), the prevalence numbers (per 105 people) were as follows: IPF 23.4 and other fibrosing ILD 102.3 of which 30.2 were PPF. In the PERSEIDS study, the prevalence in six European countries ranged between 2.8–31.0 (IPF) and 22.3–205.8 (non-IPF F-ILD) [9]. In a recent systematic literature review, overall ILD prevalence ranged from 6.3 to 76.0 in Europe and was 74.3 in the USA [14]. The collective estimated prevalence of PF-ILD ranged from 2.2 to 20 in Europe and was 28 in the USA. These estimates were considered as generally low. Correspondingly, the prevalence estimates in the PERSEIDS study, and consequently in the current study, were higher than previously reported. Similarly, prevalence numbers in a large American claims database were 117.8 for F-ILD and 70.3 for PF-ILD [15]. As in other RWE studies, a larger proportion of patients in our whole cohort and each group were men [16,17,18]. IPF patients were older at baseline, which has been the case also in other studies where ILD patients were divided in similar groups [19].

There are a few proposed criteria to define a PF-ILD—or currently PPF. According to Cottin et al. [6], patients meeting any of the following criteria within a 24-month period have experienced a disease progression: A relative decline of ≥10% in FVC; a relative decline of ≥15% in DLCO; or worsening symptoms or a worsening radiological appearance accompanied by a ≥5–<10% relative decrease in FVC. Similarly, in the INBUILD trial, to define the progressive ILD phenotype, patients were required to meet ≥1 of the following criteria in the 24 months before screening: relative decline in FVC ≥ 10% predicted; relative decline in FVC ≥5–<10% predicted and worsened respiratory symptoms; relative decline in FVC ≥5–<10% predicted and increased extent of fibrosis on HRCT; or worsened respiratory symptoms and increased extent of fibrosis on HRCT [20]. The same criteria for PPF were applied in this study. Recently, a period of 12 months was proposed to be sufficient to define the progressive nature of an ILD [4]. If this shorter time had been applied in the current study, even more patients would have probably belonged to the PPF group instead of the F-ILD group. Indicative for this was that median time since the first ILD diagnosis was, indeed, 24 months (24.5), which was the same as our criterium for the progression in the PPF group, but longer for the F-ILD group (35 months).

A total of 13–40% of patients with ILD are estimated to develop a progressive fibrosing disease [14]. In our cohort, 29.5% of fibrosing ILD patients (excl. IPF) had PPF. This was well in line with previous RWE findings from Italy (31% in a two-center study [18]), France (27.2% in a clinical cohort of the PROGRESS study [21]), and the PERSEIDS study, where a third of the F-ILD cases showed progressive behavior [9]. On the other hand, in our cohort, of the fibrosing ILD patients, a similar amount than the PPF were uncertain fibrosing patients (F-ILD-U). When database search algorithms were expanded beyond the usual definitions of PPF (lung function, imaging, symptoms) to parameters such as healthcare consumption, 47% of fibrosing ILD patients could be categorized as PF-ILD [22]. The mOS of these PF-ILD patients was 3.7 years, which could be considered as relatively low when compared to our study, where the mOS for the PPF group was not reached during the 5-year follow-up. As in our study, survival of PPF patients was slightly better than survival of IPF patients in real-world cohorts. For example, Chen et al. [23] reported a mOS of 58 and 54 months for PF-ILD and IPF, respectively. However, the mOS of the F-ILD-U group was 29.8 months and, according to our results, the disease course of this group was similar or even worse than that of IPF and PPF. It is probable that a proportion of these patients might have had a PPF. Moreso, they were older (indicating late diagnoses) than the other fibrosing patients (excl. IPF), had more UIP pattern (indicating worse prognosis), and comorbidities (indicating poor condition). This group might also contain patients from the beginning of the study period (2014) who would have been categorized in the IPF group with more modern practices. Interestingly, clearly a higher—and similar—percentage of PPF and F-ILD patients compared to IPF, and F-ILD-U patients remained in the follow-up during the 5-year period of this study. This also emphasizes the importance of prompt diagnosis and treatment initiation of these patients to maintain good quality of life and optimal healthcare resource use.

Exposure-related ILDs were the most dominant diagnosis in our cohort, both in the F-ILD and the PPF groups. This is due to frequent occupational background and related legislation of these diseases (e.g., dock yard industry in the area) and, therefore, there has been a strong interest to diagnose and follow closely these patients. Exposure-related ILDs were also the most common category in the PROGRESS study (24.2%) [22]. Unclassifiable IIP was the next frequent diagnosis and the most common in the F-ILD-U group. Particularly in this group, the unclassifiable IIP diagnosis included smoking-related restrictive pulmonary disease ILD patients who were not categorized under other diagnoses. Of the other ILD diagnoses, percentages of HP and SSc-ILD were higher in the PPF group when compared to the F-ILD group. Correspondingly, sarcoidosis and mixed CTD were more prominent in the F-ILD and F-ILD-U groups. Similarly, sarcoidosis had the lowest percent of patients who progressed (42.8%), and HP had the highest percent (74.2%) in the US database [15]. In the European countries of the PERSEIDS study, sarcoidosis was evaluated to be the most common possible reason for F-ILD (although less frequently in practice), followed by other ILDs [9]. HP, other ILDs, and unclassifiable IIP were the most common in PF-ILD. In data from Northern Finland, CTD-ILD, RA-ILD, asbestosis, and iNSIP were the following ILD diagnoses (6–14% each) after IPF [24,25]. Based on the literature, there seems to be variation in the proportions of different ILD diagnoses, which obviously originates from geography but probably also from differences in studies and diagnostics.

A similar comorbidities profile, together with gastroesophageal reflux disease, was reported in other studies [16,17,18,21]: the most common comorbidities in all groups were diabetes, cardiovascular diseases, chronic lower respiratory diseases, and sleep apnea—especially in IPF. Concerning sleep apnea, Wong et al. [19] explored how hypoxia or obstructive events caused by that disorder could result in cell damage and faster progression of fibrosis. However, although the percentage of patients with sleep apnea in IPF was higher in the current study, it was lower for also rapidly progressing PPF and F-ILD-U patients. A notable number of malignant diseases in the F-ILD-U group might have originated from cancer as a formerly diagnosed disease for these patients. Poor conditions because of the cancer might also have limited the possibility of monitoring lung function of these patients and thus excluded them from the PPF and F-ILD groups of this study.

The most common immunosuppressive medication was glucocorticoids alone or together with other immunosuppressant steroid-sparing agents. This is in line with findings from other studies where glucocorticoids were the most used treatments followed by azathioprine and mycophenolate [18,21,26]. Symptom management and older treatment practices at the beginning of the study period explained use of glucocorticoids for IPF (31.8% of the patients).

Except the IPF group, few patients had an estimation of progression of fibrosis, based on HRCT, in their records during the follow-up. In clinical practice, there has been more routine and interest to assess disease of IPF patients. Instead, radiological evaluation of other ILD patients has been more difficult and approximate. Clear signs of progression might require comparison images from a longer timespan than a couple of years. There is certainly room for improvement in using radiological imaging to evaluate progressive ILDs and, perhaps, a need to develop and adopt algorithms for the follow-up. Indeed, criteria for radiological progression are discussed in greater detail in the updated recommendations [4].

IPF patient data from the same region revealed a baseline FVC % predicted 77% and for 24% of the patients the FVC value was >90% [27]. In this study, for all ILD patients, baseline FVC % predicted was in the same range (77–80%), and in line with RWE from other countries, e.g., for IPF patients with a FVC % predicted 80% (of which 30% had FVC ≥90%) [17] and for ILD patients in general from 74 to 81% [19,21]. DLCO % predicted for our progressive groups (IPF, PPF) and F-ILD-U (56–60%) was similar to previous findings [17,19]. Regular follow-up, e.g., of CTD or RA patients, can reveal asymptomatic ILD, which explains higher baseline lung function values for PPF and F-ILD as compared to IPF. Although research using RWD provides real-world evidence, the data are often scant, making it difficult to perform analyses. Achieving enough observations proved to be a challenge in this study as well. Incomplete findings were a particular problem for FVC because progressive disease could not be determined for 145 fibrosing ILD patients. In addition, nine IPF patients had zero and 23 had one FVC measurement throughout the study period. In the previous IPF study from the same region, those who lacked FVC data had a shorter survival, which was explained by the inability to perform spirometry because of advanced disease [27]. A longitudinal decrease of 10% or more in the absolute value of FVC is characteristics for progressive ILD such as IPF and is associated with increased risk of mortality [28]. Indeed, such a decreasing tendency was observed from the baseline situation until the third follow-up year, with FVC % predicted decreasing from 77% to 66% and from 80% to 70% in the IPF and PPF groups, respectively. Remaining patients in the follow-up during the third and fourth follow-up years were probably in better physical condition as indicated by a recovery in the FVC % predicted. Maybe for the same reason, the DLCO and TLC % predicted for these groups remained constant during the follow-up, although there was also slight decrease from the baseline. Overall, lung function evolution was similar for these two groups (IPF, PPF). The lung function of F-ILD was constant throughout the follow-up and, logically, better than that of the two previously mentioned groups. In the F-ILD-U group, there were also other reasons for the poor condition of these patients than lung impairment, which explains the higher values.

Short walking test distances (<300 m) and final oxygen saturation < 85% have shown to be strong predictors of mortality [16]. PPF patients had the worst baseline 6MWT results (71.3% from theoretical distance). The same result for the IPF patients was 79%; in a previous study, the corresponding distance was 335 m [27]. In general, there were few observations regarding 6MWT during the study period and follow-up. 6MWT together with oxygen saturation assessment was conducted for specific purposes such as to evaluate eligibility for lung transplantation or the need for supplemental oxygen during exercise. The lowest SpO2% during 6MWT remained below 85% during the first years of the follow-up for the IPF, PPF, and F-ILD-U groups. Desaturation below 88% during a 6MWT has been considered as a recommendation to prescribe supplemental oxygen [28]. Although the 6MWT result was not necessarily an indication for oxygen therapy in this cohort, 14–18% of IPF and PPF patients received it yearly during the follow-up. As a comparison, long-term oxygen therapy was used for 3–9% for IPF patients in a real-world cohort [17]. The cumulative percentage of patients receiving oxygen therapy during the 5-year follow-up (in relation to patient numbers in baseline) was the following: IPF 35%, PPF 39%, F-ILD 21%, and F-ILD-U 19%. Moreover, the situation of IPF and PPF patients was similar. Correspondingly, 40% of PF-ILD patients received oxygen therapy in a seven-year Italian follow-up study [18]. In our study, there were F-ILD patients with oxygen treatment, suggesting that change in FVC as main criteria of progression does not catch all patients with progressive fibrosis.

Acute exacerbation is a serious complication of fibrosing ILDs which causes a rapid decrease in lung function and mortality [6]. In an IPF follow-up study, respiratory-related reasons for acute hospitalizations included respiratory infection (44%) and respiratory worsening (37%) [29]. Acute exacerbation was confirmed for 41% of the IPF and other ILD patients who were hospitalized because of acute respiratory symptoms [25], and incidence of acute exacerbations ranged from 3 (other ILDs) to 8 (IPF) per 100 patient years [30]. Comparing IPF and PPF in our cohort, the yearly incidence of acute hospitalizations was higher for IPF at the first year but proportions of patients during the following years were similar. About one third of the events appeared to be respiratory-related, thus resulting in an estimated yearly incidence of 5–7% of respiratory-related hospitalizations. Moreso, the disease course of IPF and PPF was similar regarding the proportion of hospitalizations. A remarkable number of F-ILD-U hospitalizations at first year was another sign of their poor condition, multimorbidity, and a probable reason for their observed higher mortality whereas F-ILD hospitalization amounts were lower than in other groups, as expected.

BMI was another studied parameter that showed a similar, slightly decreasing tendency during the follow-up in the IPF ja PPF groups. There was more variation in the follow-up BMI of F-ILD-U, indicating heterogeneity of patients in this group.

## 5. Conclusions

In a hospital setting, the number of subjects with non-IPF progressive pulmonary fibrosis exceeded the number of IPF patients. The disease course of IPF and PPF was similar. Our real-world data shows the need for prompt algorithms not only for early diagnosis of PPF but also for the follow up regimes of PPF and F-ILD. Delay in diagnosis, limitations in lung function tests, and interpretation of radiologic progression in the first 1–2 years seem to be major problems. Better healthcare resource use warrants active screening and follow-up of progressive fibrosis instead of all F-ILD.

## Figures and Tables

**Figure 1 medicina-59-00281-f001:**
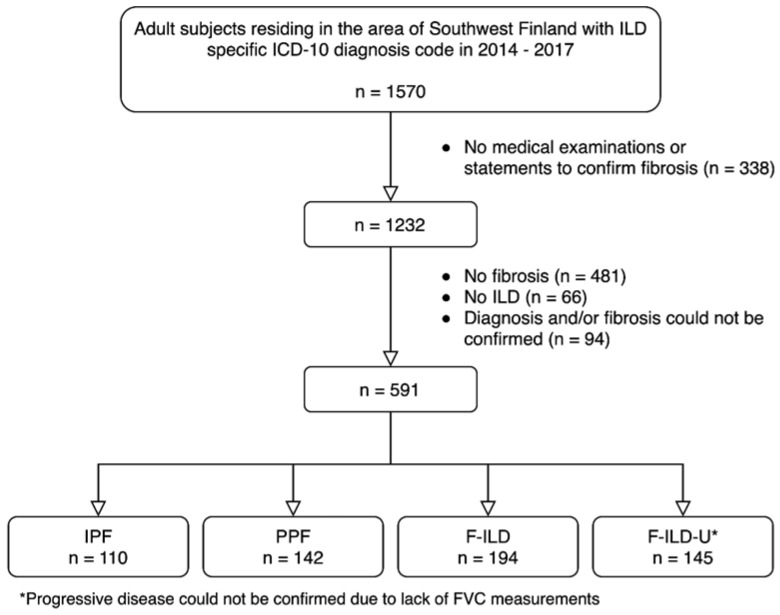
Formation of the patient cohort.

**Figure 2 medicina-59-00281-f002:**
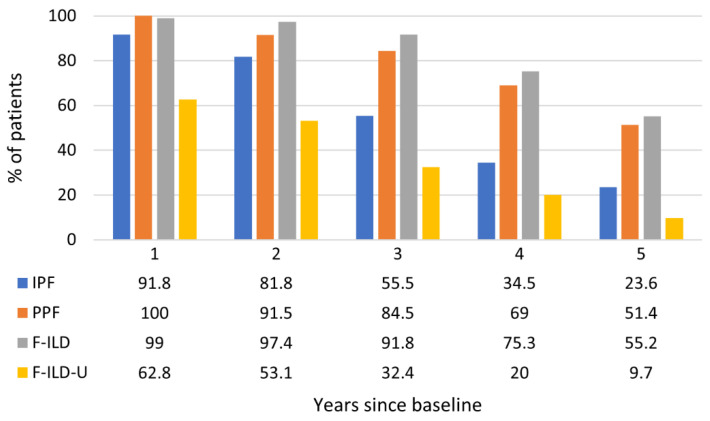
Proportions of patients in follow-up throughout the observation time.

**Figure 3 medicina-59-00281-f003:**
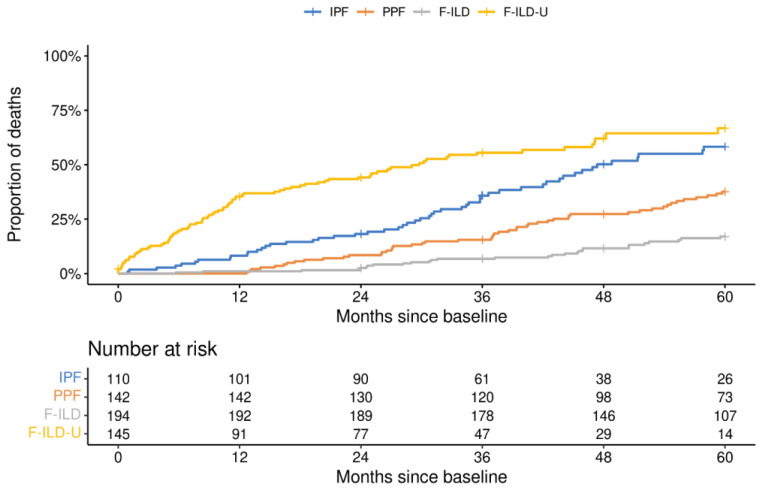
Survival of patients in the different groups.

**Figure 4 medicina-59-00281-f004:**
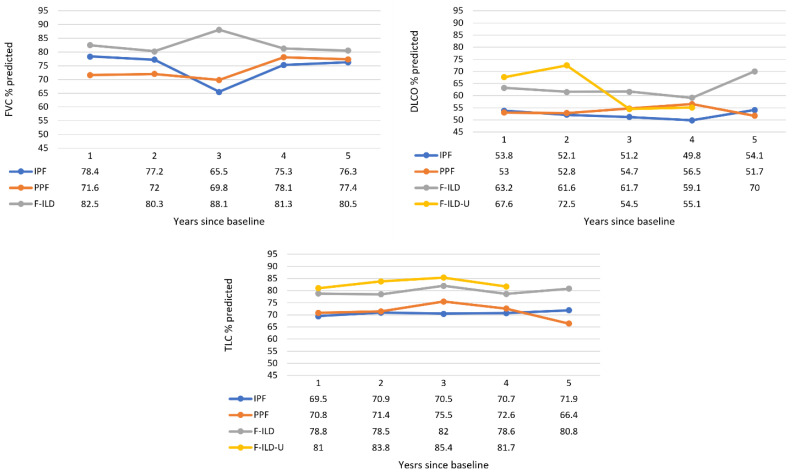
Progression of lung function measurements throughout the follow-up time. The F-ILD-U group was not included in the FVC graph as these patients had one FVC measurement available at most. Data values are medians (Q1, Q3; min–max provided in Appendix A).

**Figure 5 medicina-59-00281-f005:**
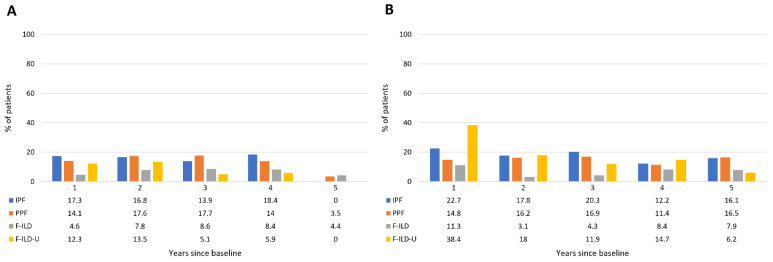
Oxygen use (**A**) and acute hospitalizations (**B**) of patients throughout the follow-up time.

**Figure 6 medicina-59-00281-f006:**
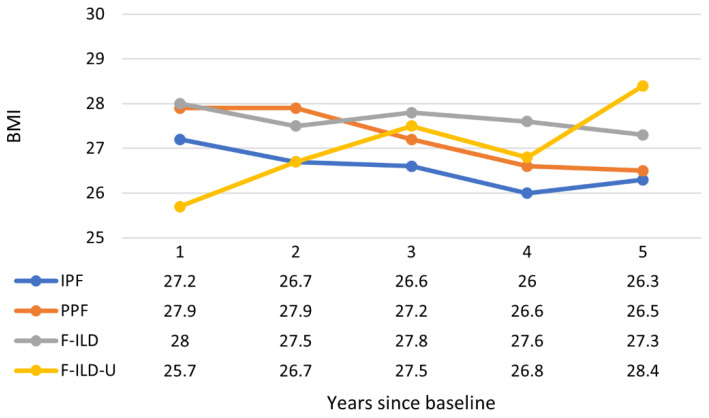
Progression of BMI throughout the follow-up time. Data values are medians (Q1, Q3; min–max provided in Appendix A).

**Table 1 medicina-59-00281-t001:** Baseline characteristics and baseline lung function of the patient groups.

Patient Group.	IPF (*n* = 110)	PPF (*n* = 142)	F-ILD (*n* = 194)	F-ILD-U (*n* = 145)
**Gender**				
Female	39 (35.5%)	49 (34.5%)	67 (34.5%)	46 (31.7%)
Male	71 (64.5%)	93 (65.5%)	127 (65.5%)	99 (68.3%)
**Age**				
Median (min-max), years	74 (54–92)	73 (24–89)	70 (24–90)	75 (20–96)
**UIP pattern**				
	103 (93.6%)	46 (32.4%)	52 (26.8%)	51 (35.2%)
**ILD diagnosis**				
IPF	110 (100.0%)	0	0	0
Exposure-related ILD	0	60 (42.3%)	75 (38.7%)	45 (31.0%)
HP	0	5 (3.5%)	3 (1.5%)	4 (2.8%)
Mixed CTD	0	2 (1.4%)	11 (5.7%)	3 (2.1%)
NSIP	0	13 (9.2%)	22 (11.3%)	14 (9.7%)
Other diffuse CTD	0	8 (5.6%)	18 (9.3%)	9 (6.2%)
RA-ILD	0	8 (5.6%)	20 (10.3%)	8 (5.5%)
Sarcoidosis	0	8 (5.6%)	18 (9.3%)	10 (6.9%)
SSc-ILD	0	11 (7.7%)	7 (3.6%)	3 (2.1%)
Unclassifiable IIP	0	27 (19.0%)	20 (10.3%)	49 (33.8%)
**Time since the first ILD diagnosis**				
Median (Q1, Q3), months	0 (0, 19.0)	24.5 (0, 87.3)	35.0 (0, 95.8)	0 (0, 53.0)
**Smoking status**				
No	31 (28.2%)	24 (16.9%)	33 (17.0%)	39 (26.9%)
Quit	25 (22.7%)	23 (16.2%)	37 (19.1%)	22 (15.2%)
Yes	18 (16.4%)	23 (16.2%)	24 (12.4%)	22 (15.2%)
Missing	36 (32.7%)	72 (50.7%)	100 (51.5%)	62 (42.8%)
**FVC % predicted**				
Median (Q1, Q3)	77.0 (61.7, 87.8)	80.0 (64.9, 93.3)	79.1 (63.5, 89.0)	78.8 (60.0, 88.2)
Missing	9 (8.2%)	0 (0%)	0 (0%)	50 (34.5%)
**DLCO % predicted**				
Median (Q1, Q3)	55.5 (45.2, 64.3)	58.7 (48.5, 71.1)	63.6 (50.0, 75.4)	59.9 (50.8, 71.6)
Missing	20 (18.2%)	36 (25.4%)	28 (14.4%)	74 (51.0%)
**TLC % predicted**				
Median (Q1, Q3)	74.3 (66.0, 82.9)	78.3 (64.0, 87.5)	78.8 (69.3, 87.7)	78.4 (69.1, 87.7)
Missing	20 (18.2%)	36 (25.4%)	28 (14.4%)	75 (51.7%)
**6 min walking test (% from theoretical distance**)				
Median (Q1, Q3)	79.0 (59.3, 103)	74.0 (53.5, 86.0)	67.0 (55.5, 100)	81.5 (60.8, 91.0)
Missing	60 (54.5%)	115 (81.0%)	162 (83.5%)	133 (91.7%)
**Lowest SpO2 %**				
Median (Q1, Q3)	88.0 (84.0, 90.8)	84.5 (76.8, 90.8)	87.0 (83.0, 91.0)	88.0 (83.8, 88.3)
Missing	60 (54.5%)	112 (78.9%)	161 (83.0%)	133 (91.7%)

**Table 2 medicina-59-00281-t002:** Baseline comorbidities and selected baseline medications of the patient groups.

Patient Group.	IPF (*n* = 110)	PPF (*n* = 142)	F-ILD (*n* = 194)	F-ILD-U (*n* = 145)
**Comorbidities**				
Any comorbidity	59 (53.6%)	80 (56.3%)	104 (53.6%)	84 (57.9%)
Diabetes	22 (20.0%)	32 (22.5%)	32 (16.5%)	22 (15.2%)
Obesity	3 (2.7%)	4 (2.8%)	4 (2.1%)	0
Sleep apnea	17 (15.5%)	11 (7.7%)	25 (12.9%)	11 (7.6%)
Cardio vascular diseases	26 (23.6%)	27 (19.0%)	26 (13.4%)	29 (20.0%)
Renal insufficiency	3 (2.7%)	8 (5.6%)	5 (2.6%)	9 (6.2%)
Chronic lower respiratory diseases	15 (13.6%)	37 (26.1%)	50 (25.8%)	29 (20.0%)
Malignant diseases	11 (10.0%)	18 (12.7%)	14 (7.2%)	31 (21.4%)
Mood and other anxiety disorders	2 (1.8%)	0	3 (1.5%)	1 (0.7%)
**Medications**				
Use of any medication (as below)	35 (31.8%)	59 (41.5%)	73 (37.6%)	57 (39.3%)
Immunosuppressants	4 (3.6%)	15 (10.6%)	20 (10.3%)	3 (2.1%)
TNF-alpha inhibitors	0	0	0	0
Selective immunosuppressants	0	5 (3.5%)	3 (1.5%)	2 (1.4%)
IL-6 inhibitors	0	0	1 (0.5%)	0
B-cell inhibitor	0	2 (1.4%)	1 (0.5%)	1 (0.7%)
Glucocorticoids	35 (31.8%)	56 (39.4%)	68 (35.1%)	54 (37.2%)

## Data Availability

Data is contained within the article and Appendix A.

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
