# Peer review of "Clinical Characteristics and Disease Course of Fibrosing Interstitial Lung Disease Patients in a Real-World Setting"

_medicina, 2023, doi:10.3390/medicina59020281_

Round 1
Reviewer 1 Report
The Study is very interesting and actual.
The disease has a progressive course and is recognized only after 65-70 years of age. This affects the already adequate treatment with antifibrotics and possibly slowing down the course of the disease
Method and results are OK as Disccusion
Literaure data has 25/30 inside last 5 years (very good)
The paper deserves to be printed
Author Response
We thank for these positive comments! About antifibrotics that the Reviewer mentions: Our study included patients with diagnosis 2014-2017 and their follow-up. The effect of antifibrotic drugs could have been seen in very minority of subjects. Pirfenidone was reimbursed in 2013 and nintedanib in 2015 in Finland. The reimbursement criteria FEV1 between 50-90% excludes patients with poor lung function and similarly no measurement in the present study. In non-IPF progressive fibrosis (PPF), the reimbursement was granted after the study period end, in 2022. We therefore assume that antifibrotics had a very minor effect on the results, and no effect among non-IPF PPF patients.
Reviewer 2 Report
Thank you for the hard work.
Please describe the exact clinical or imaging characteristics of each type of pulmonary fibrosis. How did you categorized the patients in each group?
Author Response
Thanks for these encouraging words! Characteristics and categorizing criteria of different fibrosing patients are the main focus of our study – and, therefore, the mentioned clinical, imaging, and other characteristics are the main results as well. Categorizing was based on the commonly accepted criteria for IPF and other progressive fibrosing diseases. The clinical diagnosis was based on clinical and radiological decision. First, ICD-code and specification of the disease was done by text mining. That included radiology statements, clinical diagnoses, medication, oxygen use etc. The text mined diagnoses were reviewed by two physicians together from the electronic records. In radiology we relied on radiologist’s view. The CT images had been evaluated by experienced radiologists specialized in chest radiology. They used criteria that existed at that moment. Because of the real-world setup, we did not analyze CT:s afterwards. Group IPF included patients with IPF-diagnosis and available spirometry values. Group F-ILD-U included all subjects who did not have FVC measurements but had IPF or non-IPF F-ILD, but their condition was poor for e.g. spirometry results. Group PPF included subjects that fulfilled criteria of progression on lines 94-100. Group F-ILD included those who had F-ILD, but no progression. All the above-mentioned is explained in Materials and Methods. In Discussion, we also address our groups regarding previous and current criteria for progressive disease (starting from line 325).